# Reducing Measurement Costs of Thermal Power: An Advanced MISM (Mamba with Improved SSM Embedding in MLP) Regression Model for Accurate CO_2_ Emission Accounting

**DOI:** 10.3390/s24196256

**Published:** 2024-09-27

**Authors:** Yinchu Wang, Zilong Liu, Hui Huang, Xingchuang Xiong

**Affiliations:** 1National Institute of Metrology, Beijing 100029, China; wangych@nim.ac.cn (Y.W.); liuzl@nim.ac.cn (Z.L.); 2Key Laboratory of Metrology Digitalization and Digital Metrology for State Market Regulation, Beijing 100029, China; 3State Grid Hubei Electric Power Research Institute, Wuhan 430048, China; huangh122@hb.sgcc.com.cn

**Keywords:** coal, CO_2_ emissions, carbon content as received, carbon content regression, Mamba

## Abstract

Current calculation methods for the carbon content as received (*C_ar_*) of coal rely on multiple instruments, leading to high costs for enterprises. There is a need for a cost-effective model that maintains accuracy in CO_2_ emission accounting. This study introduces an MISM model using key parameters identified through correlation and ablation analyses. An Improved State-Space Model (ISSM) and an IS-Mamba module are integrated into a Multi-Layer Perceptron (MLP) framework, enhancing information flow and regression accuracy. The MISM model demonstrates superior performance over traditional methods, reducing the Root Mean Square Error (RMSE) by 22.36% compared to MLP, and by 9.65% compared to Mamba. Using only six selected parameters, the MISM model achieves a precision of 0.27% for the discrepancy between the calculated CO_2_ emissions and the actual measurements. An ablation analysis confirms the importance of certain parameters and the effectiveness of the IS-Mamba module at improving model performance. This paper offers an innovative solution for accurate and cost-effective carbon accounting in the thermal power sector, supporting China’s carbon peaking and carbon neutrality goals.

## 1. Introduction

With the acceleration of industrialization and globalization, the impact of human activities on the environment has become increasingly prominent [1]. Among them, the issue of CO_2_ emissions has garnered significant attention due to its impact on global climate change [2,3]. China’s energy-related CO_2_ emissions, which are as high as 10.2 Gt, constitute an important part of global CO_2_ emissions [4]. In September 2020, China explicitly proposed carbon peaking and carbon neutrality goals, initiating and improving CO_2_ emission management across various industries and gradually advancing the development of a nationwide carbon trading market. The thermal power industry in China, which consumes coal and generated approximately 40% of the total CO_2_ emissions in 2022, is a key carbon-emitting sector, and was among the first to be included in China’s carbon trading market in July 2021 [5]. The Intergovernmental Panel on Climate Change (IPCC) offers two statistical calculation methods for carbon emissions from the thermal power industry. One is a monitoring method based on the Continuous Emission Monitoring System (CEMS), and the other is an accounting method based on the measured data for coal combustion [6,7]. China has conducted extensive research on the accounting method in the thermal power industry, adhering to the principle of accounting as the primary approach and monitoring as the supplementary method [8,9], ensuring fairness and equity in carbon trading in the thermal power industry.

The carbon accounting method requires enterprises to strictly adhere to the relevant standards [10,11,12,13,14,15,16,17,18,19] when collecting, analyzing, and calculating the key parameters to ensure the accuracy of the CO_2_ emission results. According to the accounting requirements specified in the Guidelines for Enterprise Greenhouse Gas Emission Accounting and Reporting for Power Generation Facilities [20], the *C_ar_* of coal burned in thermal power plants is a crucial measured parameter for carbon accounting. This parameter directly affects the accuracy of carbon accounting and guarantees an enterprise’s reasonable compliance costs. Additionally, the *C_ar_* of coal is derived from other coal quality analysis parameters and requires auxiliary parameters for data reliability verification. In addition to measuring the air-dry basis carbon content (*C_ad_*), eight other parameters need to be determined, including the total moisture (*M_ar_*), air-dry basis moisture (*M_ad_*), air-dry basis total sulfur (*S_t,ad_*), air-dry basis ash (*A_ad_*), air-dry basis volatile matter (*V_ad_*), air-dry basis hydrogen content (*H_ad_*), air-dry basis fixed carbon (*FC_ad_*), and gross calorific value (*Q_bd_*). The measurement of these parameters requires various instruments, complex procedures, and significant labor costs, resulting in significant testing costs for enterprises. If an enterprise fails to conduct an actual measurement of the elemental carbon, a default value for the carbon content per unit of calorific value, combined with the net calorific value, must be utilized to calculate the CO_2_ emissions. Although this default value has been reduced from 0.03356 tC/t (in 2019) to 0.03085 tC/t (in 2022) [21], it is still higher compared to the actual measurement results for the coal consumed by enterprises, leading to unreasonable compliance costs for enterprises.

To ensure the accuracy of carbon accounting by thermal power enterprises and reasonable compliance costs, while reducing the number of measured parameters and their corresponding instruments, thus minimizing enterprises’ measuring instruments and labor costs, numerous researchers have conducted studies on the *C_ar_* regression using only partially measured parameters. Saptoro et al. presented an empirical modeling approach for predicting the elemental composition of coal. The relationship between some of the coal parameters and the carbon content was explored [22]. Yi et al. employed a proximate analysis regression to calculate the elemental composition of coal [23]. While this method can determine the carbon content of coal, it is only applicable to a single coal type, which is unsuitable for the blended coal processing required in thermal power generation. Liu et al. compared a multivariate linear model (MLM) and maximum likelihood estimation (MLE) for analyzing the elemental composition of coal [24]. Although the MLE outperforms the MLM, its regression accuracy is limited due to its inability to effectively model the nonlinear relationship between the input parameters and carbon content. Zhu introduced commonly used empirical formulas for coal quality inspection and explained the conditions for judging data accuracy [25]. Additionally, several studies have utilized spectral data combined with Partial Least Squares (PLS) regression to calculate the carbon content [26,27,28]. Despite these methods offering faster detection speeds, they require additional analytical instruments, failing to effectively reduce the testing costs for enterprises. Saptoro et al. proposed an empirical modeling approach for coal’s elemental composition analysis, comparing MLM, principal component regression (PCR), PLS, and artificial neural networks (ANNs) [22]. The ANN method achieved good results on the test set, but the model design was overly simplified, limiting its learning capabilities. Yin et al. introduced MLP networks for predicting carbon content and calculating carbon emissions, achieving a higher regression accuracy [29]. This method enriches the neural network structure, yet it still relies on a simple stack of linear layers and activation layers, limiting its feature extraction capabilities and making it unable to properly model the relationship between the carbon content and other measured parameter sequences. With the development of sequence modeling in Natural Language Processing (NLP) [30,31], the sequence modeling capabilities of networks have been significantly enhanced, leading to the proposal of RNN [32], LSTM [33], and Transformer [34]. However, for practical applications, their complex network structures and computational mechanisms pose higher demands on the hardware, requiring substantial resource allocation. To address this, Albert et al. introduced a sequence modeling network based on the Mamba module, achieving improved computational efficiency and accuracy [35]. If this model is applied to *C_ar_* regression, further research is needed to determine the optimal selection of network input parameters. Additionally, the design of the State-Space Model (SSM) within the Mamba module fails to establish a direct connection between input and output, and the lack of consideration for information sharing between different layers can hinder the full convergence of network training.

To solve the above issues, we firstly conducted both qualitative and quantitative analyses of the relationship between *C_ar_* and other measured parameters through correlation analysis, thereby preliminarily identifying the linear correlations between *C_ar_* and some measured parameters. Further, through the ablation analysis of the measured parameters, we verified the nonlinear relationship between *M_ar_*, *M_ad_*, and *C_ar_*, and selected six measured parameters (*M_ar_*, *M_ad_*, *V_ad_*, *A_ad_*, *FC_ad_*, and *Q_bd_*) for subsequent regression calculations of *C_ar_*. Based on the Multi-Layer Perceptron (MLP) framework, we introduced the IS-Mamba module with the Improved State-Space Model (ISSM), incorporating multiple skip connections to share information between layers, thereby constructing the MISM regression model. After that, we compared the performance of different methods on public datasets to demonstrate the effectiveness of the MISM regression model. Finally, we validated the reliability of the measured parameter reduction and the effectiveness of the regression model design through testing experiments.

## 2. Correlation Analysis

In this section, a correlation analysis was performed on the interrelationships among nine measured parameters, including *M_ar_*, *M_ad_*, *S_t,ad_*, *V_ad_*, *A_ad_*, *H_ad_*, *FC_ad_*, *Q_bd_*, and *C_ar_*. The relationship between *C_ar_* and the remaining eight parameters to facilitate subsequent regression analysis was the focus. The complete analysis process included coal combustion parameter collection and preprocessing, scatter matrix analysis, and correlation coefficient calculation.

### 2.1. Collection and Preprocessing of Coal Parameters

We collected actual measurement data from a thermal power plant in Hubei, China, from 1 September 2022 to 31 August 2023. The power plant has two generating units, and daily coal quality analysis and parameter measurement are required for the coal consumed by both units. Due to shutdowns and routine maintenance, a total of 687 data records were collected. Each data record includes nine parameters: *M_ar_*, *M_ad_*, *S_t,ad_*, *V_ad_*, *A_ad_*, *H_ad_*, *FC_ad_*, *Q_bd_*, and *C_ar_*. After eliminating missing and abnormal values, a total of 529 data records were compiled. All data were normalized using the min-max scaling method for subsequent analysis.

### 2.2. Scatter Matrix Analysis

To visually illustrate the relationships between various coal combustion parameters for qualitative analysis, we plotted a scatter plot matrix, as shown in Figure 1. The trends in the last row of the figure indicate that *A_ad_*, *FC_ad_*, and *Q_bd_* have significant linear relationships with *C_ar_*, with *Q_bd_* and *FC_ad_* exhibiting positive correlations and *A_ad_* exhibiting a negative correlation. Additionally, *V_ad_* and *H_ad_* also display a certain linear relationship with *C_ar_*, which is positive. However, *M_ar_*, *M_ad_*, and *S_t,ad_* do not show clear linear relationships with *C_ar_*.

### 2.3. Correlation Coefficient Calculation

We further quantitatively described the correlations identified in the qualitative analysis of Section 2.2 using the Pearson correlation coefficient (PCC). The resulting PCC matrix is presented in Figure 2. The correlation coefficients between *FC_ad_*, *Q_bd_*, and *C_ar_* are above 0.8, indicating a very strong positive linear correlation. The correlation coefficient between *A_ad_* and *C_ar_* is less than −0.8, indicating a very strong negative linear correlation. Meanwhile, the correlation coefficients between *V_ad_*, *H_ad_*, and *C_ar_* fall between 0.6 and 0.8, suggesting a relatively strong positive linear correlation. The importance ranking of these five parameters, from highest to lowest, is *Q_bd_*, *A_ad_*, *FC_ad_*, *V_ad_*, and *H_ad_*. The correlation coefficients between *M_ar_*, *M_ad_*, *S_t,ad_*, and *C_ar_* are all between −0.2 and 0.2, indicating a very weak linear correlation. However, this does not exclude the possibility of nonlinear correlations between these parameters and *C_ar_*. To further analyze the importance of each parameter, this study conducted a parameter ablation analysis to investigate the influence of each parameter on *C_ar_*.

## 3. Parameter Ablation Analysis

In order to further investigate the importance of the other parameters on *C_ar_*, we took *C_ar_* as the output and the other parameters as the inputs, and used the MISM method to obtain the following calculation formula:(1)$$C_{ar}=0.5630\times Q_{bd}+0.4566\times M_{ad}-0.3689\times M_{ar}+0.2521\times {FC}_{bd}+\;0.1365\times V_{ad}+0.0906\times A_{ad}+0.0227\times H_{ad}-0.0112\times S_{t,ad},$$

The absolute values of the coefficients of each parameter in the formula reflect the importance of each parameter, and the order of importance from highest to lowest is *Q_bd_*, *M_ad_*, *M_ar_*, *FC_ad_*, *V_ad_*, *A_ad_*, *H_ad_*, and *S_t,ad_*.

To further validate the importance of each parameter, we conducted a parameter ablation analysis by eliminating one parameter at a time and substituting its default value of 0.5 into Equation (1), while keeping other parameters unchanged. The comparison of *C_ar_*’s calculated results is presented in Table 1. Three comparison metrics, MAE, RMSE, and MAPE, are considered. In Table 1, the first row labeled “baseline” represents the test error without eliminating any parameters, while the remaining rows indicate the test error when each parameter is set to its default value. Among the parameters, *S_t,ad_* had the smallest impact on the calculation results when set to its default value, with a decrease of 1.01% in MAE and 0.73% in RMSE. The second-smallest impact was observed for *H_ad_*, with an increase of 2.23% in MAE and 1.17% in RMSE. Notably, setting *Q_bd_* to its default value had a particularly significant impact on the calculation results, leading to a significant increase of 166.40% in MAE and 147.95% in RMSE. Based on the absolute size of ΔMAE, we ranked the importance of the parameters in descending order: *M_ad_*, *Q_bd_*, *M_ar_*, *FC_ad_*, *A_ad_*, *V_ad_*, *H_ad_*, and *S_t,ad_*.

Based on the aforementioned analysis, we further conducted a step-by-step reduction of parameters and labeled the corresponding measuring sensors that could be reduced as a result of this reduction, as shown in Table 2. In this table, “×” indicates that the parameter has been reduced (set to the default value of 0.5), and the instruments are labeled with the actual models used in the power plant in Hubei. Specifically, 5E-AS3200B (Automatic Coulomb Sulfur Analyzer produced by Changsha Kaiyuan Instruments Co., Ltd., Changsha, China) is a sulfur analyzer used for determining *S_t,ad_*, while 5E-CHN2200 (C/H/N Elemental Analyzer produced by Changsha Kaiyuan Instruments Co., Ltd., Changsha, China) is a carbon–hydrogen–nitrogen analyzer for measuring *C_ad_* and *H_ad_*. 5E-MW6510 (Automatic Moisture Analyzer produced by Changsha Kaiyuan Instruments Co., Ltd., Changsha, China) serves as an industrial analyzer for determining *M_ad_*, *A_ad_*, and *V_ad_*, with *FC_ad_* being calculated based on these parameters. 5E-MAG670 (Proximate Analyzer produced by Changsha Kaiyuan Instruments Co., Ltd., Changsha, China) is a moisture meter for Mar measurement, and 5E-C5500 (Automatic Calorimeter produced by Changsha Kaiyuan Instruments Co., Ltd., Changsha, China) is a calorimeter for *Q_bd_* determination. From the results of the reduction, it is evident that after setting the default values for *S_t,ad_* and *H_ad_*, the MAE and RMSE only increased by 1.01% (from 4.94 to 4.99) and 0.29% (from 6.82 to 6.84), allowing for the reduction of two actual measuring instruments. However, when *A_ad_* and *V_ad_* were further set to their default values, although MAE and RMSE decreased by 1.01% (from 4.94 to 4.89) and 4.69% (from 6.82 to 6.50), no further reduction in the types of actual measuring instruments was achieved. Additionally, upon assigning default values to other parameters, the errors increased significantly, resulting in a decrease in calculation accuracy.

## 4. MISM Regression Model for Calculating *C_ar_*

While the parameters that could be reduced were identified as described in Section 3, the relationship between *M_ar_* or *M_ad_* and *C_ar_* is not linearly correlated. The MLM method is more suitable for modeling linear relationships and has limited capabilities for potential nonlinear relationships. Therefore, on the basis of MLP with nonlinear capabilities, this paper proposes an IS-Mamba module to further improve the modeling performance. An Improved State-Space Model (ISSM) is combined with Mamba to construct the IS-Mamba module. Furthermore, skip connections are utilized for feature sharing between different layers. Ultimately, an MISM regression model is constructed for calculating *C_ar_*.

### 4.1. Improved State-Space Model (ISSM)

In the traditional Mamba structure, the calculation model of SSM is
(2)$$\left\{\begin{array}{c} h_{k}=\bar{A}h_{k-1}+\bar{B}x_{k}\\ y_{k}=Ch_{k} \end{array}\right.$$

To achieve a selective SSM, $\bar{A}$, $\bar{B,}$ and *C* are all obtained based on *x_k_* [36]. The output *y_k_* of this model is only related to the current state *h_k_*, making the effect of *x_k_* on *y_k_* indirect. As the number of network layers increases, the original information in *x_k_* undergoes multiple linear transformations and activations, resulting in a certain loss of information, which is not conducive to inter-layer information transmission. Therefore, this paper proposes an Improved State-Space Model (ISSM), which introduces a direct transfer matrix *D* to directly transfer the information of *x_k_* to *y_k_*. The computational diagram of this model is shown in Figure 3. To maintain the selectivity of the model, the direct transfer matrix *D* is calculated from the input *x_k_* through a linear layer and a SoftPlus layer.

### 4.2. IS-Mamba Design

We integrate the ISSM designed in Section 4.1 with the Mamba module [36] to form the basic feature mapping module IS-Mamba for the *C_ar_* regression model. The model structure is shown in Figure 4. For the input features, after normalization through the RMSnorm layer, two branches are used for feature mapping. The first branch, after being mapped through a linear layer, integrated by a convolutional layer, and activated by the SiLU(·) function, serves as the input to the ISSM model to further obtain the mapped features *f*_ISSM_. The expression of the SiLU(·) function is
(3)$$SiLU\left(x\right)=\frac{x}{1+e^{-x}}$$

The second branch undergoes linear mapping and then directly passes through the SiLU(·) function to obtain a gated result *f*_gate_ with feature importance selection. The output of the IS-Mamba module is then obtained by multiplying *f*_ISSM_ with *f*_gate_ and further mapping through a linear layer.

### 4.3. MISM Regression Model

The structure of the MISM regression network designed in this paper is shown in Figure 5. Based on a two-layer MLP network, we incorporate the designed IS-Mamba into the network and add two skip connections to better utilize low-level information directly in the higher layers. The input *x* = {*M_ar_*, *M_ad_*, *S_t,ad_*, *V_ad_*, *A_ad_*, *H_ad_*, *FC_ad_*, *Q_bd_*} to the MISM is mapped by IS-Mamba. Then, the mapped features are directly added to the input *x* to obtain the summed feature *f*_add_.
(4)$$f_{add}=IS\_Mamba\left(x\right)+x$$

The feature *f*_add_ is integrated and mapped through a linear layer and a LeakyReLU activation layer. The integrated feature is then further mapped through the second IS-Mamba block and added to *f*_add_, allowing the input and the features mapped by the first IS-Mamba block to directly affect this layer. Finally, the feature is integrated and mapped into a single value through a linear layer and a LeakyReLU layer, serving as the regression result for the normalized *C_ar_*. The LeakyReLU(·) function used in the MISM is defined as
(5)$$LeakyReLU\left(x\right)=\left\{\begin{array}{cc} x & x>0\\ \alpha x & x\leq 0 \end{array}\right.$$
where α is a fixed parameter. In the experiments, we set α as 0.01.

### 4.4. Model Details

The model details of ISSM, IS-Mamba, and MISM are shown in Table 3, where *bs* represents the batch size. For ISSM, Table 3 presents the dimensions of $\bar{A}$, $\bar{B}$, *C*, *D*, and *h_k_*, where *L* = 8 and *s* = 128 represent the state-space size. For IS-Mamba and MISM, Table 3 shows the output dimension of each layer, where *M* = 16. It is especially worth noting that, in the conv layer, the convolution kernel size is set to 1 × 3, with padding set to 1.

## 5. Experiments and Analysis

In the experiments, we firstly introduced the settings of the training parameters and the dataset details. Then, different models were tested to compare the performance of *C_ar_* regression. After that, an ablation study was adopted to demonstrate the effectiveness of each optimization. Finally, we quantitatively analyzed the feasibility of the MISM regression model for CO_2_ emission calculation.

### 5.1. Datasets and Implementation Details

#### 5.1.1. Datasets

House Prices dataset. The House Prices dataset on Kaggle is a standard house price prediction dataset commonly used for regression problems. This dataset contains 1460 instances (i.e., houses), each with 79 feature variables that can be used to predict house prices. The features include continuous numerical features, discrete label features, and possible missing values, with the target variable being the sales price of a property. We standardized all numerical features and filled in missing values with zero values. For discrete values, we used a one-hot encoding method to fill in. If a discrete feature had two different discrete values, the two attributes were processed as two-dimensional [0,1] and [1,0], respectively. When there were three different discrete values, they corresponded to [0,0,1], [0,1,0], and [1,0,0]. We divided the data with real housing prices into K parts in the order of their IDs and conducted K-fold cross validation, with K = 5 in the experiment.

Diabetes dataset. In Python’s sklearn database, the Diabetes dataset is a commonly used regression dataset, which is used to predict the disease progress of patients one year later. There are 442 patient records in the dataset, and 10 physiological indicators of each patient are used as characteristics to predict the quantitative measurement of disease progress (one year later). The features in the dataset have been mean-centered and scaled to standard deviation, and can be directly used for model training and testing. In the experiment, the data were divided into K parts in the order of their IDs and subjected to K-fold cross validation. In the experiment, K was set as 5.

Coal dataset. We collected and organized the Coal dataset. For the introduction of this dataset, refer to Section 2.1. We partitioned the dataset based on odd and even months, where the odd-month data served as the training set, comprising a total of 272 entries. The even-month data served as the testing set, containing 257 entries. The distribution of *C_ar_* in the training and testing sets is depicted in Figure 6.

#### 5.1.2. Implementation Details

The MISM regression model was implemented on PyTorch-1.13.0. We conducted model training and testing on NVIDIA GeForce RTX 3070Ti (Graphics Processing Unit produced from NVIDIA Corporation, Santa Clara, CA, USA) with 8 GB of memory. The process of using the MISM regression model to regress *C_ar_* is shown in Table 4. We used the Adam [36] optimizer to tune the model parameters, with *β*_1_ = 0.9 and *β*_2_ = 0.999. The strategy and other parameter settings are shown in Table 5. The warm-up strategy [37] was adopted in the coal experiments. The first 64 epochs were the warm-up stage, and the following 192 epochs were the learning rate decay stage. The learning rate *lr* = {*lr_epoch_*|*epoch* = 1, 2, 3,..., 256} is calculated as follows:(6)$${lr}_{epoch}=\left\{\begin{array}{cc} \frac{epoch}{64}\times {lr}_{max} & epoch\leq 64\\ {{lr}_{epoch-1}}^{1.001} & 64<epoch\leq 256 \end{array}\right.$$
where *lr_max_* is the max. learning rate, which is set to 0.005. 

In order to reduce the sensitivity of the network to outliers during training, we used L1 loss as the training loss of the network:(7)$$loss=\sum_{i=1}^{n}\left|y_{i}-{y_{i}}^{\prime }\right|$$
where *y_i_* represents the normalized measured *C_ar_* and *y_i_′* represents the regression results of MISM. 

We quantitatively compared the proposed method with existing methods using the seven evaluation indicators in Section 5.2. These evaluation indicators are

Mean Absolute Error (MAE)


(8)
$$MAE\left(y,y^{\prime }\right)=\frac{1}{n}\sum_{i=1}^{n}\left|y-y^{\prime }\right|,$$


2.Root Mean Square Error (RMSE)


(9)
$$RMSE\left(y,y^{\prime }\right)=\sqrt{\frac{1}{n}\sum_{i=1}^{n}{\left(y_{i}-{y_{i}}^{\prime }\right)}^{2}},$$


3.Mean Absolute Percentage Error (MAPE)


(10)
$$MAPE\left(y,y^{\prime }\right)=\frac{1}{n}\sum_{i=1}^{n}\frac{\left|y_{i}-{y_{i}}^{\prime }\right|}{y_{i}}\times 100\%,$$


4.Coefficient of Determination (R^2^)


(11)
$$R^{2}\left(y,y^{\prime }\right)=1-\frac{\sum_{i=1}^{n}{\left(y_{i}-{y_{i}}^{\prime }\right)}^{2}}{\sum_{i=1}^{n}{\left(y_{i}-\bar{y}\right)}^{2}},$$


5.Pearson Correlation Coefficient (PCC)


(12)
$$PCC\left(y,y^{\prime }\right)=\frac{cov\left(y,y^{\prime }\right)}{\sigma_{y}\sigma_{y^{\prime }}},$$


6.Concordance Correlation Coefficient (CCC)


(13)
$$CCC\left(y,y^{\prime }\right)=\frac{2PCC\left(y,y^{\prime }\right)\sigma_{y}\sigma_{y^{\prime }}}{\sigma_{y}^{2}+\sigma_{y^{\prime }}^{2}+{\left(\mu_{y}-\mu_{y^{\prime }}\right)}^{2}},$$


7.Explained Variance (Evar)

(14)$$Evar\left(y,y^{\prime }\right)=1-\frac{\sigma_{y-y^{\prime }}^{2}}{\sigma_{y}^{2}},$$
where *σ_x_* represents the standard deviation of *x*, and *μ_x_* represents the mean of *x*. The values of MAE, RMSE, and MAPE are all greater than 0, where smaller values indicate better prediction performance. The values of R^2^, PCC, CCC, and Evar are all between 0 and 1. 

### 5.2. Experiments on House Prices Dataset

To verify the effectiveness of our proposed method, we compare the MISM model with existing models on the House Prices dataset. The quantitative comparison results are shown in Table 6. MISM achieved the best performance in all seven metrics. Compared to the MLP, MISM showed a reduction in MAE by 77.06% (from 2.79 to 0.64) and a reduction in RMSE by 76.47% (from 4.42 to 1.04). When compared to the original Mamba, MISM achieved a reduction in MAE by 39.05% (from 1.05 to 0.64) and a reduction in RMSE by 31.13% (from 1.51 to 1.04). To provide an intuitive comparison of the performance of various methods, we constructed a radar chart, as depicted in Figure 7. To ensure that a larger shaded area indicates superior performance, we normalized MAE, RMSE, and MAPE by dividing each by its respective maximum value obtained across different methodologies. Subsequently, we subtracted this normalized value from 1.0. Utilizing this adjusted value in conjunction with four additional metrics, we constructed the radar chart. Based on the comparative analysis, the MISM method exhibits a marginal advantage over the Mamba method, yet it demonstrates significant superiority over other existing approaches.

### 5.3. Experiments on Diabetes Dataset

To further verify the effectiveness of our proposed method, we compared the MISM model with existing models on the Diabetes dataset. The quantitative comparison results are shown in Table 7. MISM achieved the best performance in all seven metrics. Compared to the Mamba, MISM achieved a reduction in MAE by 14.39% (from 4.17 to 3.57) and a reduction in RMSE by 11.24% (from 5.34 to 4.74). The qualitative comparison results are shown in Figure 8. The construction of the radar chart adheres to the methodology outlined in Section 5.2. MISM also achieved the best performance.

### 5.4. Experiments on Coal Dataset

In this subsection, two approaches were adopted for the model regression testing. In the first approach, all eight input parameters were measured values, and the quantitative comparison results are presented in Table 8. Among the seven comparison metrics, MISM achieved the best performance in five metrics and the second-best performance in two metrics. Compared to the MLP, MISM showed a reduction in MAE by 23.11% (from 6.10 to 4.69) and a reduction in RMSE by 19.95% (from 8.02 to 6.42). When compared to other neural network-based methods, MISM exhibited superior regression performance. In comparison with the original Mamba, MISM achieved a reduction in MAE by 12.66% (from 5.37 to 4.69) and a reduction in RMSE by 11.45% (from 7.25 to 6.42).

In the second approach, default values were used for *S_t,ad_* and *H_ad_*, while the remaining parameters were measured values. The quantitative comparison results are presented in Table 9. MISM achieved the best performance in all seven metrics. Compared to the MLP, MISM showed a reduction in MAE by 27.71% (from 6.17 to 4.46) and a reduction in RMSE by 22.36% (from 7.96 to 6.18). When compared to the original Mamba, MISM achieved a reduction in MAE by 12.20% (from 5.08 to 4.46) and a reduction in RMSE by 9.65% (from 6.84 to 6.18).

A comparative analysis of Table 8 and Table 9 reveals that when *S_t,ad_* and *H_ad_* adopt default values, the errors exhibit the most significant increase in the SVR algorithm, with an MAE increase of 14.55% (from 7.01 to 8.03) and an RMSE increase of 9.96% (from 9.34 to 10.27). Conversely, the RNN demonstrates the most significant reduction in errors, exhibiting a 7.90% decrease in MAE (from 5.95 to 5.48) and a 4.61% decrease in RMSE (from 7.60 to 7.25). For the MISM model, MAE reduces by 4.90% (from 4.69 to 4.46), while RMSE decreases by 3.74% (from 6.42 to 6.18). Therefore, the proposed MISM structure in this study exhibits a certain improvement in accuracy when *S_t,ad_* and *H_ad_* are set to default values. In addition, considering the changes in RMSE, MLP, RNN, Mamba, and MISM are more suitable for using six key parameters for carbon content regression. After reducing to six key parameters, their RMSE decreased by 0.75% (from 8.02 to 7.69), 4.61% (from 7.60 to 7.25), 5.66% (from 7.25 to 6.84), and 3.74% (from 6.42 to 6.18), respectively. MLM, SVR, RF, and LSTM are more suitable for using eight parameters for carbon content regression. After reducing to six key parameters, their RMSE increased by 0.29% (from 6.82 to 6.84), 9.96% (from 9.34 to 10.27), 1.12% (from 8.96 to 9.06), and 1.65% (from 8.49 to 8.64), respectively.

### 5.5. Ablation Analysis

To validate the effectiveness of the proposed modules, we conducted ablation experiments on the collected data. These ablation experiments employed two comparative methods. The first method involved using all eight parameters as the measured values, and the results are presented in Table 10. When skip connections were added to the base MLP in Method A, the MAE decreased by 16.72% (from 6.10 to 5.08). However, the inclusion of an additional Mamba in Method B, which builds upon Method A, did not optimize the performance, leading to an 8.27% increase in MAE (from 5.08 to 5.50). Conversely, Method C, which incorporates two Mambas on the basis of Method A, achieved an improvement, with a 5.31% decrease in MAE (from 5.08 to 4.81). After applying three Mambas in Method D, the MAE increased by 0.98% compared to Method A (from 5.08 to 5.13) and by 6.65% compared to Method C (from 4.81 to 5.13). Therefore, the utilization of two Mambas yielded better performance. Subsequently, we replaced these two blocks with the proposed IS-Mamba to form the MISM network structure. Compared to Method C, the IS-Mamba module further reduced the MAE by 2.49% (from 4.81 to 4.69).

The second approach involved using default values for *S_t,ad_* and *H_ad_* among the eight input parameters, while the remaining parameters were measured for the ablation experiment. The results are presented in Table 11. Adding skip connections to the base MLP (Method A) reduced the MAE by 18.48% (from 6.17 to 5.03). Further, the inclusion of two Mambas (Method C) led to a 6.16% decrease in MAE (from 5.03 to 4.72). After replacing Mamba with IS-Mamba, the MAE was further reduced by 5.51% (from 4.72 to 4.46).

The comparative analysis of the ablation experiments conducted in the two mentioned approaches validates the effectiveness of the MISM module and its structural improvements. Furthermore, in practical applications, the use of default values for *S_t,ad_* and *H_ad_* can still ensure the accuracy of *C_ar_* calculations.

### 5.6. CO_2_ Emission Calculation

We set *S_t,ad_* and *H_ad_* to 0.5. The denormalization results of the MISM results are the calculated *C_ar_*s. The *C_ar_*s values are used to calculate CO_2_ emissions using the carbon accounting formula:(15)$$E_{combustion}=FC\times C_{ar}\times OF\times \frac{44}{12}$$
where *E*_combustion_ represents the CO_2_ emissions of coal combustion. *FC* represents the coal consumption. *OF* represents the carbon oxidation rate, which is taken as 99% [20].

We calculated the carbon emissions for four months of 2022 and ten months of 2023 in the collected data according to formula (15), and the comparison with the calculation results based on the measured *C_ar_* is shown in Table 12. Compared with the measurement method, the maximum error using MISM regression is −0.91% (in Jun. 2023), the minimum error is only 0.01% (in November 2022), and the overall error is only −0.27%.

We also conducted a daily visualization comparison of the data from the fourteen months presented in Table 12. Since the thermal power plant consists of two units, separate statistics were performed for each unit, and some of the results are displayed in Figure 9. The visualization results indicate that the carbon accounting results obtained using the proposed MISM regression model after reducing the number of parameters are highly consistent with those calculated using all measured parameters.

## 6. Discussion

This study presents a significant advancement in the field of carbon accounting for thermal power enterprises by introducing an MISM regression model, which accurately estimates the carbon content as received (*C_ar_*) using only six key parameters. Our research demonstrates that this model not only maintains high precision in regression but also substantially reduces the need for multiple measuring instruments, thereby minimizing labor and equipment costs for businesses.

One of the pivotal contributions of this research is the potential cost savings it offers to the industry. By employing the MISM model, which requires fewer parameters for accurate *C_ar_* estimation, enterprises can eliminate the need for specific measuring instruments. Specifically, this study indicates that the 5E-AS3200B sulfur analyzer and the 5E-CHN2200 carbon–hydrogen–nitrogen analyzer can be reduced, with their purchase prices being approximately CNY 1.5 million and 3 million, respectively.

Considering the scale of the industry in China, with 2257 thermal power enterprises participating in the carbon market in 2024, the MISM model could lead to a collective saving of nearly CNY 10.16 billion in instrument procurement costs. This figure underscores the substantial financial impact of our model on the industry. Moreover, the MISM model’s applicability extends beyond the thermal power sector, offering potential savings to other industries with carbon measurement needs, such as steel and aluminum smelting.

In addition to the direct cost savings from reduced instrument use, this study also considers the significant labor costs associated with operation and maintenance of these instruments. With an average salary of CNY 6948 per month for coal testers in China, eliminating two types of instruments could save an individual enterprise CNY 13,896 per month in labor costs. When scaled across the national landscape of 2257 enterprises, this reduction amounts to an annual saving of CNY 3.76 billion in labor costs.

Looking forward, the MISM model opens avenues for future research and development. The potential integration of the MISM model with other carbon accounting frameworks and its application in various industrial sectors present exciting opportunities. Furthermore, the model’s efficiency and accuracy invite exploration into its use for real-time monitoring and dynamic adjustment of carbon emissions, which could be critical for enterprises aiming to achieve carbon neutrality.

## 7. Conclusions

This study introduces the MISM, a regression model that simplifies CO_2_ emission accounting in the thermal power sector by minimizing the need for extensive parameter measurement. By integrating an Improved State-Space Model (ISSM) and the IS-Mamba module, the MISM improves the estimation of carbon content as received (*C_ar_*), which is essential for accurate IPCC CO_2_ emission accounting. The model achieves a notable reduction in RMSE, outperforming MLP by 22.36% and Mamba by 9.65%, with an emission calculation error of only 0.27%. The MISM model balance the accuracy and cost-effectiveness of carbon accounting. Future research directions could consider applying the MISM model to various datasets and evaluating its adaptability in different industries.

## Figures and Tables

**Figure 1 sensors-24-06256-f001:**
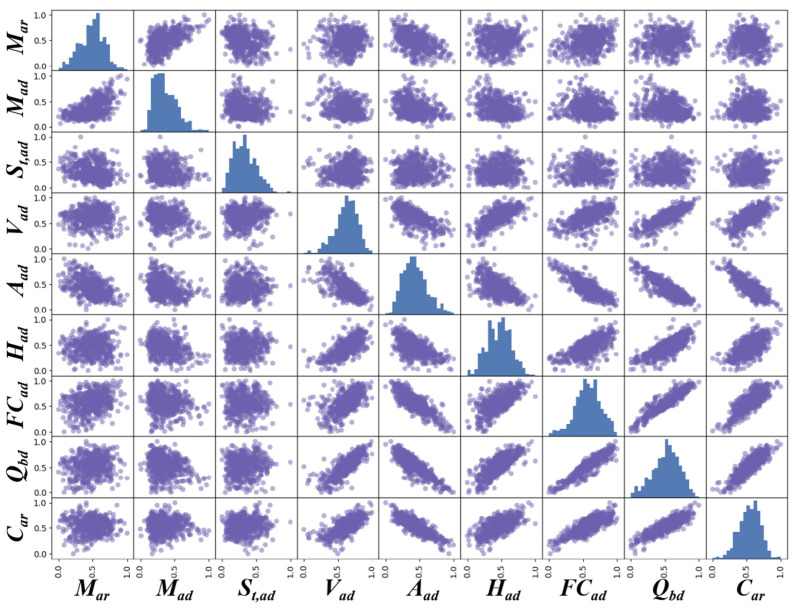
Scatter matrix for visualizing the relationships between all measured parameters. (The blue bars represent the distribution histogram of the variables).

**Figure 2 sensors-24-06256-f002:**
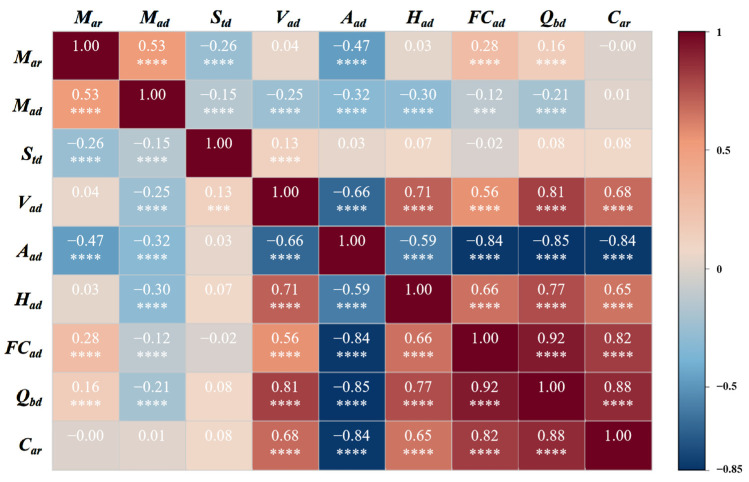
Correlation coefficient matrix of coal parameters. (*** represents *p* < 0.05; **** represents *p* < 0.001).

**Figure 3 sensors-24-06256-f003:**
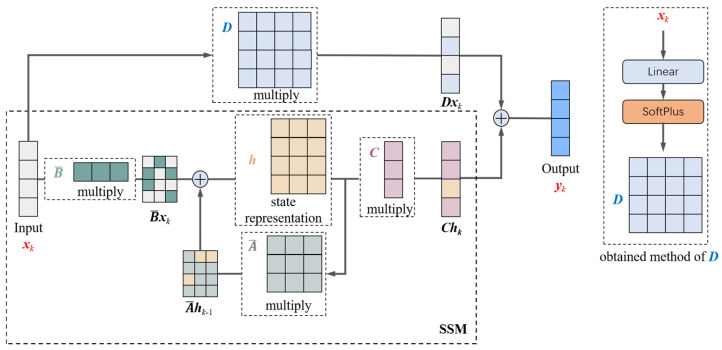
The calculation process of ISSM.

**Figure 4 sensors-24-06256-f004:**
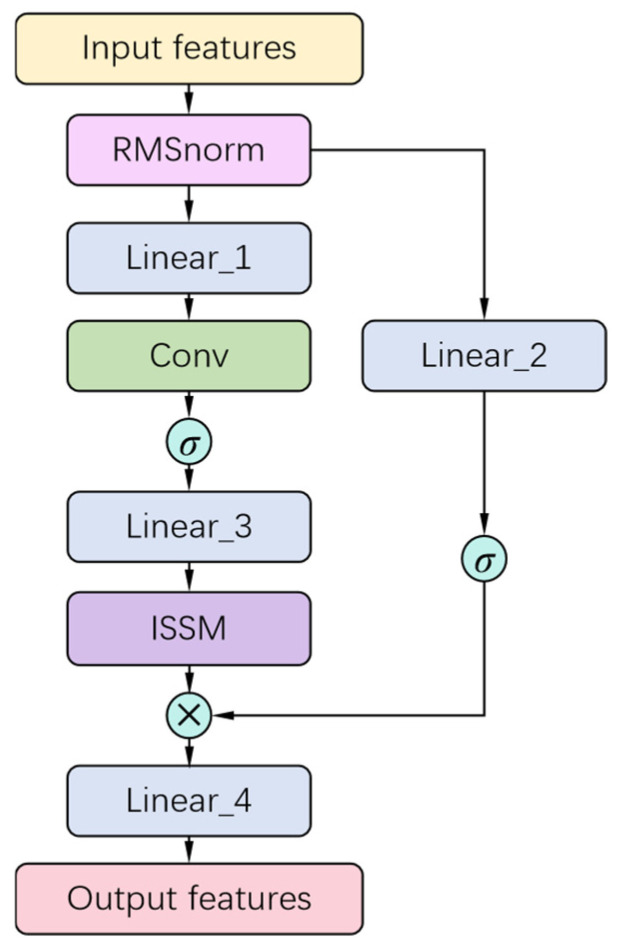
The structure of IS-Mamba.

**Figure 5 sensors-24-06256-f005:**
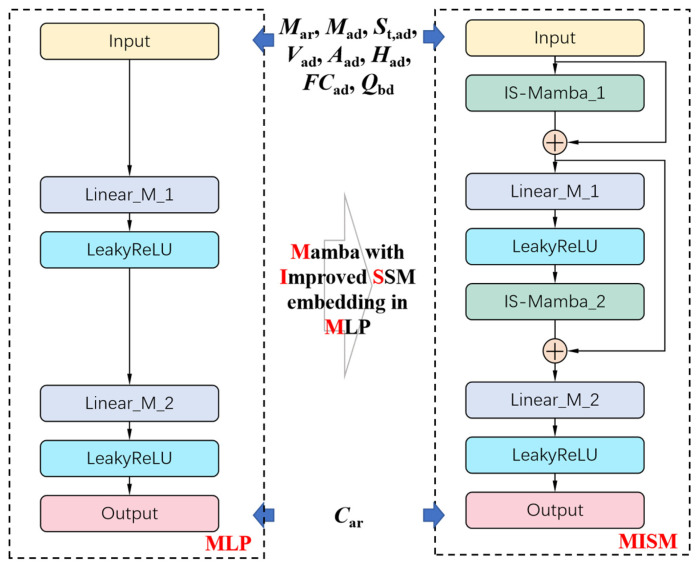
The structure of MISM regression model for calculating *C_ar_*.

**Figure 6 sensors-24-06256-f006:**
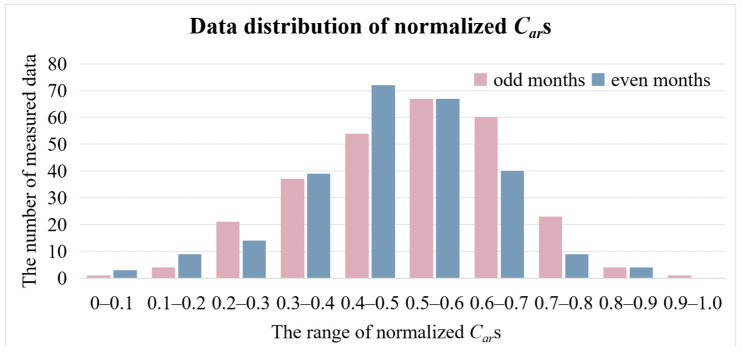
Data distribution of normalized *C_ar_*s in the dataset.

**Figure 7 sensors-24-06256-f007:**
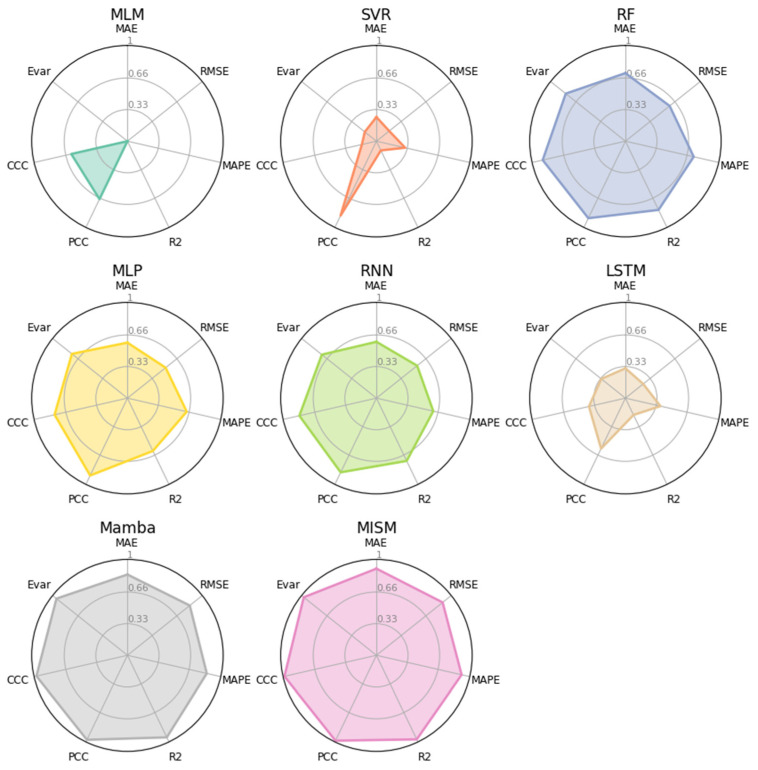
Qualitative comparison of different methods on House Prices dataset.

**Figure 8 sensors-24-06256-f008:**
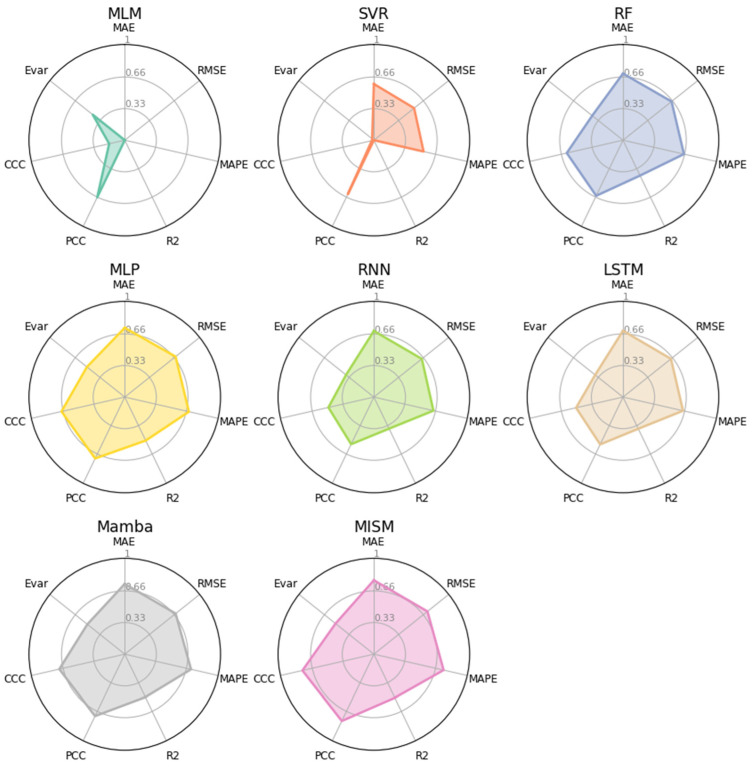
Qualitative comparison of different methods on the Diabetes dataset.

**Figure 9 sensors-24-06256-f009:**
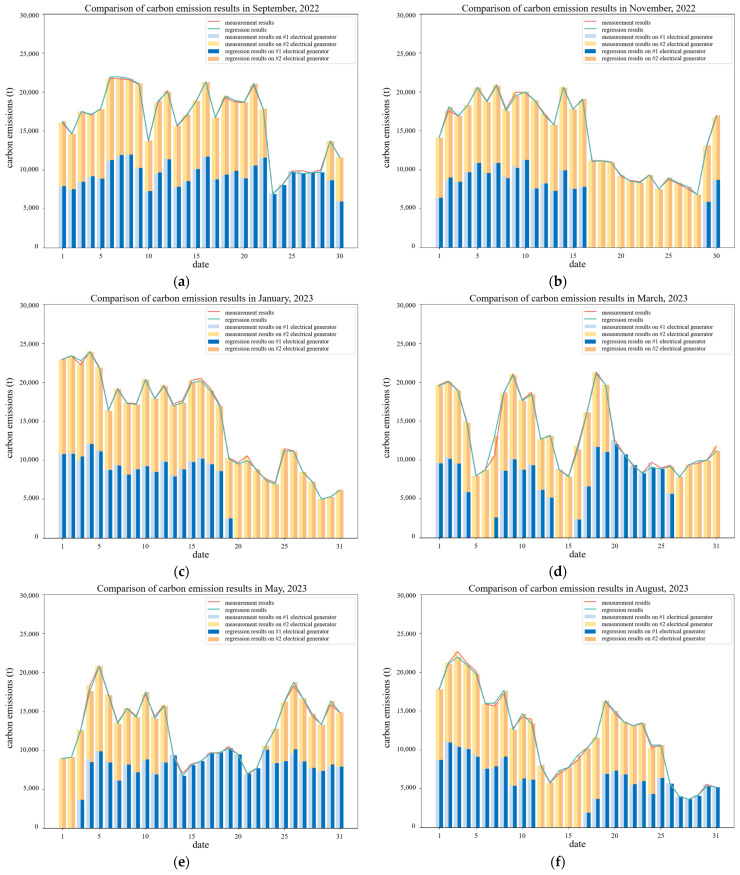
The comparison of MISM regression results (*S_t,ad_* = 0.5, *H_ad_* = 0.5) and all measured parameters for CO_2_ emission calculation: (**a**) the results of September 2022; (**b**) the results of November 2022; (**c**) the results of January 2023; (**d**) the results of March 2023; (**e**) the results of May 2023; and (**f**) the results of July 2023.

**Table 1 sensors-24-06256-t001:** The impact of each parameter on *C_ar_* calculation. (Bold indicates the best).

	MAE (×10^3^)	RMSE (×10^3^)	MAPE (×10^3^)	ΔMAE (%)	ΔRMSE (%)	ΔMAPE (%)
baseline	4.94	6.82	0.91	-	-	-
*M_ar_* = 0.5	9.07	11.13	1.65	83.60	63.20	0.74
*M_ad_* = 0.5	14.54	17.27	2.65	194.33	153.23	1.74
*S_t,ad_* = 0.5	**4.89**	**6.77**	**0.90**	**−1.01**	**−0.73**	**−0.01**
*V_ad_* = 0.5	5.37	6.88	0.98	8.70	0.88	0.07
*A_ad_* = 0.5	5.48	7.27	1.00	10.93	6.60	0.09
*H_ad_* = 0.5	5.05	6.90	0.93	2.23	1.17	0.02
*FC_ad_* = 0.5	7.38	9.88	1.36	49.39	44.87	0.45
*Q_bd_* = 0.5	13.16	16.91	2.41	166.40	147.95	1.50

**Table 2 sensors-24-06256-t002:** The impact of gradually reducing parameters and the instruments that can be reduced.

	0	1	2	3	4	5	6	7	8
*S_t,ad_*		×	×	×	×	×	×	×	×
*H_ad_*			×	×	×	×	×	×	×
*A_ad_*				×	×	×	×	×	×
*V_ad_*					×	×	×	×	×
*FC_ad_*						×	×	×	×
*M_ar_*							×	×	×
*M_ad_*								×	×
*Q_bd_*									×
MAE (×10^3^)	4.94	4.89	4.99	5.40	4.89	7.87	9.81	10.33	17.03
RMSE (×10^3^)	6.82	6.77	6.84	7.20	6.50	10.09	12.44	13.27	22.12
MAPE (%)	0.91	0.90	0.91	0.99	0.89	1.43	1.77	1.90	3.14
R^2^	0.8963	0.8979	0.8957	0.8847	0.9061	0.77	0.6551	0.6075	-
PCC	0.9558	0.9550	0.9552	0.9550	0.9527	0.9153	0.8144	0.8647	-
CCC	0.9446	0.9454	0.9435	0.9434	0.9489	0.8425	0.7906	0.7593	-
Evar	0.9127	0.9111	0.9105	0.9095	0.9062	0.7743	0.6631	0.7256	-
Instrument	-	5E-AS3200B	5E-CHN2200	-	-	-	5E-MW6510	5E-MAG6700	5E-C5500

**Table 3 sensors-24-06256-t003:** Parameter settings of MISM.

**ISSM**	
Input	*bs* × *L*
$\bar{A}$	*bs* × *L* × *s*
$\bar{B}$	*bs* × *L* × *s*
C	*bs* × *s*
*D*	*bs* × *L*
*h_k_*	*bs* × *L* × *s*
Output	*bs* × *L*
**IS-Mamba**	
Input	*bs* × *L*
Linear_1	*bs* × *L* × *M*
conv	*bs* × *L* × *M*
Linear_2	*bs* × *M*
Linear_3	*bs* × *M*
ISSM	*bs* × *M*
Linear_4	*bs* × *L*
Output	*bs* × *L*
**MISM**	
Input	*bs* × *L*
IS_Mamba_1	*bs* × *L*
Linear_M_1	*bs* × *L*
IS_Mamba_2	*bs* × *L*
Linear_M_2	*bs* × *L*
Output	*bs* × 1

**Table 4 sensors-24-06256-t004:** The process of using MISM model to regress and calculate *C_ar_*.

Input	coal parameters *x*
Step 1	Min-Max Normalization Min-Max(*x*) → *x_norm*
Step 2	model initialization
Step 3	MISM calculation
	IS-Mamba_1(*x_norm*) → *features* Add(*x_norm*, *features*) → *x_add* Linear_M_1(*x_add*) → *x_linear* LeakyReLU(*x_linear*) → *x_act* IS-Mamba_2(*x_act*) → *features* Add(*x_add*, *features*) → *x_add* Linear_M_2(*x_add*) → *x_linear* LeakyReLU(*x_linear*) → *MISM_features*
Output	Denormalization
	Denorm(*MISM_features*) → *C_ar_*

**Table 5 sensors-24-06256-t005:** The strategy and other parameter settings for experiments on different datasets.

	House Prices	Diabetes	Coal
Warm-Up	×	×	√
bs	26	6	16
lr	0.00002	0.00065	*lr_max_ =* 0.005
weight decay	0.00005	0.000007	0.00005
iters	200	200	256
K-Fold Cross Validation	√	√	×

**Table 6 sensors-24-06256-t006:** The comparison results on the House Prices dataset. (Bold indicates the best).

	MAE (×10^−4^)	RMSE (×10^−4^)	MAPE (%)	R^2^	PCC	CCC	Evar
MLM [24]	6.64	9.02	40.34	-	0.67	0.60	-
SVR [38]	4.97	7.48	27.93	0.11	0.86	0.16	0.16
RF [39]	4.57	3.69	10.83	0.80	0.89	0.89	0.80
MLP [29]	2.79	4.42	14.72	0.61	0.90	0.78	0.74
RNN [32]	2.73	4.11	15.79	0.73	0.86	0.83	0.73
LSTM [33]	4.57	6.88	25.21	0.19	0.59	0.39	0.32
Mamba [35]	1.05	1.51	6.06	0.95	0.98	0.97	0.95
MISM	**0.64**	**1.04**	**3.53**	**0.97**	**0.99**	**0.99**	**0.97**

**Table 7 sensors-24-06256-t007:** The comparison results on the Diabetes dataset. (Bold indicates the best).

	MAE (×10^−1^)	RMSE (×10^−1^)	MAPE (%)	R^2^	PCC	CCC	Evar
MLM [24]	15.61	16.63	120.44	-	0.66	0.17	0.43
SVR [38]	6.44	7.70	56.35	-	0.63	0.02	0.02
RF [39]	4.74	5.85	41.83	0.41	0.64	0.60	0.41
MLP [29]	4.31	5.35	37.70	0.50	0.71	0.68	0.50
RNN [32]	4.83	5.99	43.63	0.37	0.55	0.49	0.37
LSTM [33]	4.79	5.98	43.02	0.37	0.55	0.50	0.37
Mamba [35]	4.17	5.34	35.00	0.49	0.71	0.70	0.50
MISM	**3.57**	**4.74**	**30.54**	**0.50**	**0.77**	**0.76**	**0.51**

**Table 8 sensors-24-06256-t008:** The comparison of different methods (eight measured parameters as input). (Bold indicates the best. Underlined indicates the second best).

	MAE (×10^3^)	RMSE (×10^3^)	MAPE (%)	R^2^	PCC	CCC	Evar
MLM [24]	4.94	6.82	0.91	0.8963	**0.9558**	0.9446	**0.9127**
SVR [38]	7.01	9.34	1.29	0.8056	0.9325	0.8891	0.8602
RF [39]	7.08	8.96	1.29	0.8208	0.9211	0.9081	0.8468
MLP [29]	6.10	8.02	1.11	0.8564	0.9278	0.9179	0.8569
RNN [32]	5.95	7.60	1.08	0.8711	0.9376	0.9271	0.8751
LSTM [33]	6.53	8.49	1.19	0.8393	0.9239	0.9026	0.8396
Mamba [35]	5.37	7.25	0.98	0.8827	0.9396	0.9375	0.8829
MISM	**4.69**	**6.42**	**0.85**	**0.9080**	0.9540	**0.9533**	0.9092

**Table 9 sensors-24-06256-t009:** The comparison of different methods (*S_t,ad_* and *H_ad_* are set as 0.5, and the others are measured parameters). (Bold indicates the best. Underlined indicates the second best).

	MAE (×10^3^)	RMSE (×10^3^)	MAPE (%)	R^2^	PCC	CCC	Evar
MLM [24]	4.99	6.84	0.91	0.8957	0.9552	0.9435	0.9105
SVR [38]	8.03	10.27	1.49	0.7645	0.9499	0.8568	0.8589
RF [39]	7.26	9.06	1.32	0.8171	0.9199	0.9055	0.8452
MLP [29]	6.17	7.96	1.11	0.8587	0.9385	0.9162	0.8674
RNN [32]	5.48	7.25	1.00	0.8827	0.9522	0.9290	0.8852
LSTM [33]	6.60	8.63	1.21	0.8340	0.9375	0.8926	0.8383
Mamba [35]	5.08	6.84	0.93	0.8955	0.9486	0.9413	0.8955
MISM	**4.46**	**6.18**	**0.81**	**0.9149**	**0.9565**	**0.9557**	**0.9149**

**Table 10 sensors-24-06256-t010:** The ablation experiments with all eight measured parameters as input. (Bold indicates the best. “√” indicates the adoption of a particular module).

	MLP	Mamba	A	B	C	D	MISM
MLP	√		√	√	√	√	√
skip connections			√	√	√	√	√
Mamba × 1				√			
Mamba × 2		√			√		√
Mamba × 3						√	
IS-Mamba × 2							√
MAE	6.10	5.37	5.08	5.50	4.81	5.13	**4.69**
RMSE	8.02	7.25	6.90	7.14	6.45	6.75	**6.42**

**Table 11 sensors-24-06256-t011:** *S_t,ad_* and *H_ad_* are taken as default values and the other parameters are measured for ablation study. (Bold indicates the best. “√” indicates the adoption of a particular module).

	MLP	Mamba	A	B	C	D	MISM
MLP	√		√	√	√	√	√
skip connections			√	√	√	√	√
Mamba × 1				√			
Mamba × 2		√			√		√
Mamba × 3						√	
IS-Mamba × 2							√
MAE	6.17	5.08	5.03	5.25	4.72	5.08	**4.46**
RMSE	7.96	6.84	6.80	6.90	6.40	6.67	**6.18**

**Table 12 sensors-24-06256-t012:** The comparison between MISM results and actual measurement results.

Month	MISM (tCO_2_)	Measurement (tCO_2_)	MAPE (%)
September 2022	486,444.04	485,380.15	0.22
October 2022	389,993.22	389,106.39	0.23
November 2022	431,299.54	431,275.39	0.01
December 2022	453,086.83	454,370.59	−0.28
January 2023	459,769.47	463,191.43	−0.74
February 2023	501,989.43	505,267.05	−0.65
March 2023	404,665.92	405,723.63	−0.26
April 2023	284,092.72	286,180.73	−0.73
May 2023	397,055.89	397,682.09	−0.16
June 2023	350,878.94	351,232.17	−0.10
July 2023	490,627.54	495,131.89	−0.91
August 2023	372,076.01	371,236.99	0.23
September 2023	330,377.13	330,676.25	−0.09
October 2023	30,082.55	30,332.60	−0.82
Total	5,382,439.23	5,396,787.35	−0.27

## Data Availability

The original contributions presented in the study are included in the article, further inquiries can be directed to the corresponding author.

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
