# Peer review of "Reducing Measurement Costs of Thermal Power: An Advanced MISM (Mamba with Improved SSM Embedding in MLP) Regression Model for Accurate CO2 Emission Accounting"

_sensors, 2024, doi:10.3390/s24196256_

Round 1

Reviewer 1 Report

Comments and Suggestions for Authors

This article focuses on the reduction of cost of carbon accounting in the thermal power industry and proposes a MISM model based on multi-layer perceptron (MLP) and Mamba, aiming to select some key parameters to regress coal's received carbon content (Car), thereby reducing detection and compliance costs while ensuring accurate carbon accounting. Overall, the topic of this article has practical significance, with clear research ideas, detailed data, and reasonable methods. However, there is still room for improvement in the following aspects:

1. In the abstract and conclusion sections, the author stated that six key parameters were selected for modeling through correlation analysis and ablation analysis, and the method was scientifically rigorous. However, in the experimental section, eight parameters were inputted into the model. Perhaps my understanding is incorrect. Could the author further elaborate on the specific process and criteria for parameter selection to increase transparency and reproducibility.

2. The experiment was validated using three datasets (House Prices, Diabetes, and Coal). However, regarding the partitioning and preprocessing process of these datasets, it is hoped that the author can provide further clarification.

3. The experimental results were analyzed in detail, and the superiority of the MISM model in various indicators was visually demonstrated through tables and radar charts. However, for some outliers, null values, or fluctuations in the experimental results, their possible causes and solutions can be further discussed.

4. The upper part of Figures 4 and 5 is obscured. The clarity of Figure 6 needs to be improved.

Comments on the Quality of English Language

Minor editing of English language required

Reviewer 2 Report

Comments and Suggestions for Authors

The authors introduce a MISM regression model that uses six selected key parameters identified through correlation and ablation analyses. They developing an improved state-space model, integrated into an MLP framework, which utilizes skip-connections to optimize information flow between layers, thereby enhancing the regression of Car. This study has a certain significance for improving the accuracy of carbon emission estimation. I have only the following suggestions,

1. The fonts in Figure 1, Figures 6 and 9 are too small. It is recommended to adjust the fonts to a suitable size.

2. Only the correlation coefficient is marked in Figure 2, but whether it is statistically significant is not marked.

3. If Figure 1 and Figure 2 use the same metrics for people analysis, it is recommended to keep only one.

Comments on the Quality of English Language

No

Reviewer 3 Report

Comments and Suggestions for Authors

This study introduces the MISM regression model and compares various fitting methods based on the House Prices dataset, Diabetes dataset, and Coal dataset to demonstrate the precision superiority of the MISM. The study's workload and the rigor of data analysis are commendable, yet there are several areas that require improvement:

In the introduction, it is stated that "eight other parameters need to be determined." Please provide the corresponding references to support this assertion.

In section 5.1.2, the authors mention the implementation of the MISM model using PyTorch. It is recommended that the authors elaborate on the implementation process in greater detail or present it in the form of pseudocode.

The authors mention that "The measurement of these parameters requires various instruments, complex procedures, and significant labor costs," yet this issue seems to have been overlooked in the subsequent data analysis.

It is suggested that a "Discussion" section be added to further synthesize and articulate the contributions of this paper.

Reviewer 4 Report

Comments and Suggestions for Authors

The comments are included in the attached report
